# Microstructure and Wear Resistance of Laser-Clad Ni–Cu–Mo–W–Si Coatings on a Cu–Cr–Zr Alloy

**DOI:** 10.3390/ma16010284

**Published:** 2022-12-28

**Authors:** Xiaojun Zhao, Qi Zhong, Pengyuan Zhai, Pengyu Fan, Ruiling Wu, Jianxiao Fang, Yuxiang Xiao, Yuxiang Jiang, Sainan Liu, Wei Li

**Affiliations:** 1School of Materials Science and Engineering, Central South University, Changsha 410083, China; 2Key Laboratory of Non-Ferrous Metal Materials Science and Engineering, Ministry of Education, Central South University, Changsha 410083, China; 3New Technology Promotion Institute of China Ordnance Industries, Beijing 100089, China; 4School of Resource Processing and Bioengineering, Central South University, Changsha 410083, China; 5Powder Metallurgy Institute, Central South University, Changsha 410083, China

**Keywords:** laser cladding, Cu–Cr–Zr alloys, mental silicides, wear resistance

## Abstract

To improve the wear resistance of high-strength and high-conductivity Cu–Cr–Zr alloys in high-speed and heavy load friction environments, coatings including Ni–Cu, Ni–Cu-10(W,Si), Ni–Cu–10(Mo,W,Si), and Ni–Cu–15(Mo,W,Si) (with an atomic ratio of Mo,W to Si of 1:2) were prepared using coaxial powder-feeding laser cladding technology. The microstructure and wear performance of coatings were chiefly investigated. The results revealed that (Mo,W)Si_2_ and MoNiSi phases are found in the Ni–Cu–10(Mo,W,Si) and Ni–Cu–15(Mo,W,Si) coating. WSi_2_ phases are found in the Ni–Cu–10(W,Si) coating. The degree of grain refinement in Ni–Cu–10(Mo,W,Si) was greater than that of the Ni–Cu–10(W,Si) coating after the effect of Mo. The excellent wear resistance and micro-hardness of the Ni–Cu–15(Mo,W,Si) coating were attributed to the increase in its dispersion phase, which were approximately 34.72 mg/km and 428 HV, 27.1% and 590% higher than the Cu–Cr–Zr substrate, respectively. The existence of silicide plays an important role in grain refinement due to the promotion of nucleation and the inhibition of grain growth. In addition, the wear mechanism transformed from adhesive wear in the Ni–Cu coating with no silicides to abrasive wear in the Ni–Cu–15(Mo,W,Si) coating with high levels of silicides.

## 1. Introduction

Cu–Cr–Zr alloys have numerous advantages and are thus widely used in different industries, including aerospace, mechanical, electrical, and microelectronics [1,2,3,4]. However, surfaces of copper alloys are susceptible to severe wear under harsh conditions due to low hardness [5,6,7]. Surface modification technology prolongs the service life of workpieces by improving their surface performance [8,9,10]. Laser cladding technology has been widely applied due to its numerous advantages, such as rapid heating and cooling rates, low dilution rates, small heat-affected zones, and metallurgical bonding with the substrate. Laser cladding technology also provides a new direction for the surface modification of copper alloys [11,12]. Coaxial powder-feeding laser cladding technology can greatly improve deposition efficiency and surface accuracy and reduce the dilution rate [13].

The difficulties of cladding on copper alloy surfaces, due to their poor wettability and coefficients of thermal expansion, can be reduced by employing the strong wettability between Ni and Cu [14,15,16,17]. Liu et al. [18] designed Ni–Cu/Ni gradient coatings on copper alloy surfaces, which exhibited a minor improvement in wear and good metallurgical bonding. Co-based and Ni-based composite coatings are prepared through the Ni-based transition layer on the copper alloy. The wear resistance of the substrate was improved by 20-fold and 2-fold, respectively [19,20]. In addition, composite coatings with excellent wear resistance can be prepared on copper alloys by employing Ni-based solid solution powder as the binder phase and a ceramic phase as the reinforcing phase [21,22,23].

The in situ synthesis of the reinforcing phase can provide better interfacial bonding properties between the enhanced phase and binder phase [24,25,26]. Metal silicide ceramics, represented by Cr_13_Ni_5_Si_2_, MoSi_2_, and WSi_2_, have dual metal and ceramic properties, high melting points, hardness, and wear resistance [27,28,29]. The in situ synthesis of metal silicide can also be applied to cladding on the surface of copper alloys. This can not only prepare wear-resistant coatings but also reduce the differences in the thermal expansion coefficient. The addition of silicide provides a hard phase to improve the wear resistance of the composites, which has been explored elsewhere [30]. K. Wang prepared Cr_13_Ni_5_Si_2_ base metal silicide composite coatings on copper alloy surfaces by laser cladding; the wear properties of the coating at room temperature were close to plasma-sprayed NiCr/Cr_3_C_2_ cemented carbide coatings [31].

In this study, coatings of Ni–Cu, Ni–Cu–10(W,Si), Ni–Cu–10(Mo,W,Si), and Ni–Cu–15(Mo,W,Si), were prepared on a Cu–Cr–Zr alloy surface by employing coaxial powder-feeding laser cladding technology. The advantages of the metal silicides and the laser cladding were utilized to improve the wear resistance of coatings. The structures and phase composition of the coatings were studied; micro-hardness and friction wear properties of the coatings were tested; and the formation and wear mechanisms were determined.

## 2. Experimental Methods

### 2.1. Raw Materials

The substrate used in the experiment was a Cu–Cr–Zr alloy (Cu–Cr0.88–Zr0.18, wt.%) exhibiting good mechanical properties and electrical conductivity. Specimens were cut into 60 × 30 × 8 mm by EDM wire-cutting technology and then polished with 200, 400, and 600 mesh grit sandpaper. Subsequently, the specimens were ultrasonically cleaned in an alcohol solution for 15 min to eliminate oil and impurities.

The average particle size of the Ni, Cu, and Si powder was 35 µm. The average particle size of Mo and W was 10 µm. All powders were analytically pure. According to the preliminary experimental results, the Ni–Cu–Mo–W–Si coating prepared by prefabrication exhibited better performance. Thus, the self-designed composition of coatings containing Ni, Cu, Mo, W, and Si is shown in Table 1. To ensure uniformity and sphericity of the powders, weighed powders were mixed in a GH-5L powder mixer at alternate times of 12 h and speeds of 50 r/min. Before laser cladding, the mixed powders and cleaned samples were dried in a vacuum oven at 120 °C.

### 2.2. Coating Preparation

Powders with different ratios were used to fabricate coatings on the Cu–Cr–Zr alloy through laser cladding technology. Mixed powders were placed in the laser cladding feeder(ZKZM-10000W, Xi’an, China); then, the coatings were prepared in an argon atmosphere by employing ZKZM-10000 W rapid laser cladding equipment(ZKZM-10000W, Xi’an, China) with coaxial powder feeding. Based on optimization parameters determined in a prior experiment, the processing parameters of the laser cladding were as follows: laser power of 4500 W, laser scan speed of 30 mm/s, laser beam diameter of 5 mm, powder feeding rate of 18 g/min, and defocus level of 16 mm. The preparation process, cladding parameters, and performance test are given in Figure 1.

### 2.3. Microstructural Characterization

X-ray diffraction (XRD, Smart Lab 3 KW, Tokyo, Japan) using Cu K_α_ radiation at 30 kV and 100 mA was performed to analyze the phases of coatings. The diffraction angle (2θ) for sample analysis varied from 20° to 110°. Microstructural and elemental compositions of the coatings were determined by scanning electron microscopy (SEM, TESCAN-MIRA4-LMH, Brno, The Czech Republic) and energy-dispersive X-ray spectroscopy (EDS, Ultim Max 40), respectively. The specimens were etched with a mixture of distilled water (10 mL) + alcohol (10 mL) + Fe (NO_3_)_3_ (2.7 g) solution, and then cleaned in ethanol. The distribution of elements in the coating was observed with an electron probe microanalyzer EPMA, JEOL JXA-8230, JEOL Ltd., Tokyo, Japan) equipped with a spectrometer (WDS).

### 2.4. Performance Test

The Vickers microhardness was determined on a polished surface by employing a Shimadzu HMV-2T Vickers hardness tester (Shimadzu HMV-2T, Tokyo, Japan) at a load of 10 N for a dwell time of 15 s by ASTM C1327. Points were tested in the vertical direction of the coating interface. The distance of the measured point was 100 μm in the vertical direction and 400 μm in the horizontal direction. The mean value was taken as the microhardness of the coating at this position to obtain the hardness data. Hardness curves between the substrate and the coatings were derived.

Tribological wear tests were performed via HT 1000 type tribological wear apparatus with the ball–disk contact method following ISO 18535. Test parameters were as follows: friction ball of GCr15 with a diameter of 6 mm; load of 2000 g; rotation speed of 400 r/min; circular running track of Φ5 mm; and time of 30 min. To further elucidate the wear mechanisms, morphologies of the contact surfaces were characterized by SEM (SEM, TESCAN-MIRA4-LMH) and ultra-deep 3D microscopy after wear testing. Weight changes of the specimens were measured using an electric balance with a resolution of 0.1 mg to examine the wear rate. Equation (1) was used to calculate the wear rate.
(1)$$R=\frac{\left(m_{0}-m\right)}{2n\pi rt}$$
where m refers to the weight after wear, $m_{0}$ refers to the weight before wear, n refers to the rotational speed, t refers to the time of wear, r refers to radius of wear, and R refers to the wear rate.

## 3. Results and Discussion

### 3.1. Phase Analysis of Coatings

XRD patterns of the coatings are shown in Figure 2a,b. The Ni–Cu coating was primarily composed of a Ni–Cu solid solution because Ni and Cu with the same crystal structure can form an infinite solid solution. New phases such as WSi_2_ and Ni_x_Si_y_ (stable phase NiSi_2_ and sub-stable phases NiSi, Ni_2_Si, and Ni_31_Si_12_) appeared in the Ni–Cu–10(W,Si) coating with the addition of W and Si elements. Furthermore, different phases, such as Mo_5_Si_3_, MoSi_2_, and MoNiSi, are also observed in the Ni–Cu-Mo-W-Si coating (including Ni–Cu–10(Mo,W,Si) and Ni–Cu–15(Mo,W,Si) coatings) after the addition of Mo. The main diffraction peaks were shifted to smaller angles in the XRD pattern of Ni–Cu–Mo–W–Si coatings. The original interplanar spacing was altered because Ni_x_Si_y_ compounds preferentially form clusters and are dispersed in the lattice of the Ni–Cu solid solution. Mo and W atoms with large atom radii penetrate the Ni–Cu and Cu–Ni–Si solid solution lattices, causing lattice distortion, as shown in Figure 2c. The lattice distortion increases the interplanar spacing, and the diffraction angle becomes smaller as per the Bragg equation (2dsinθ = nλ) [32]. Therefore, the main diffraction peak shifted to a smaller angle.

Gibbs’s free energy (ΔG) for possible reactions in the molten pool during laser cladding was calculated using HSC Chemistry (Figure 3). Except for solid solution phases, Reactions (R1)–(R5) may occur in the molten pool during the preparation of laser cladding.
2Ni + Si = Ni_2_Si(R1)
Ni + 2Si = NiSi_2_(R2)
Ni_2_Si + Si = 2NiSi(R3)
Ni_2_Si + 3Si = 2NiSi_2_(R4)
Ni + Si = NiSi(R5)

The ΔG of Reactions (R1)–(R5) is negative, indicating that all reactions could spontaneously proceed. Due to the rapid heating and cooling characteristics of laser cladding, some of the sub-stable phases represented by NiSi and Ni_2_Si were not sufficiently converted to the NiSi_2_ stable phase in time. Therefore, NiSi compounds in the coating existed in the form of complex composite phase Ni_x_Si_y_, as shown in Figure 2b. Reactions (R6)–(R12) may occur with Mo, W, and Si elements.
5Mo + 3Si = Mo_5_Si_3_(R6)
Mo_5_Si_3_ + 7Si = 5MoSi_2_(R7)
Mo_3_Si + 5Si = 3MoSi_2_(R8)
5W + 3Si = W5Si_3_(R9)
W_5_Si_3_ + 7Si = 5WSi_2_(R10)
Mo + 2Si = MoSi_2_(R11)
W + 2Si = WSi_2_(R12)

The ΔG of all Reactions (R6)–(R12) is negative, implying that Mo_5_Si_3_ and W_5_Si_3_ and MoSi_2_ and WSi_2_ are produced [33,34]. Some Mo_5_Si_3_ and W_5_Si_3_ are not fully converted to MoSi_2_ and WSi_2_, due to the addition of only 2.9–4.3 wt.% Si.

### 3.2. Microstructure of Coatings

Cross-sectional and bonding zone structure characteristics of the Ni–Cu coating are shown in Figure 4a,b. The substrate has a uniform microstructure and grain distribution. The Ni–Cu coating consisted of Ni–Cu solid solution (point 1) with a dense structure and free cracks near the interface bond. Microscopic morphologies of the Ni–Cu–10(W,Si) coating structure are shown in Figure 4c–f. The metallurgical bonding zone is observed. The coating is scattered with white-dotted precipitation phases (point 5) as well as larger W particles (point 2). White-dotted precipitation phases are embedded in crystals and diffusely distributed in the coating. The crystalline structure of the coating from the bottom to the top is large grain, slender cellular crystals, and dendrite crystals, sequentially (Figure 4d–f).

Based on the XRD diffraction pattern and elemental content shown in Table 2, white-dotted points are WSi_2_ (point 5). Cellular crystals were mainly composed of Ni–Cu and Ni_x_Si_y_ (point 4) because Si atoms in the crystal were embedded in the Ni–Cu lattice and formed a stable phase of NiSi_2_. Due to grain boundaries with high energy and chemical instability, sub-stable phases (such as Ni_31_Si_12_ and Ni_2_Si) are segregated at the grain boundaries, resulting in the enrichment of Si (point 3).

Figure 5 shows the microstructural morphology of the Ni–Cu–10(Mo,W,Si) and Ni–Cu–15(Mo,W,Si) coatings. The crystal structure of the Ni–Cu–10(Mo,W,Si) coating was more refined than that of the Ni–Cu–10(W,Si) coating. In addition, the grain refinement of the Ni–Cu–15(Mo,W,Si) coating was better than that of the Ni–Cu–10(Mo,W,Si) coating. Figure 5a–c exhibits agglomeration in the Ni–Cu–10(Mo,W,Si) coating. Relatively homogeneous dispersion was observed in the Ni–Cu–15(Mo,W,Si) coatings, as shown in Figure 5d–f. A considerable number of white and grey phases (point 5 and point 6) was also observed.

According to the microstructural and elemental contents detailed in Table 3, large particles of the coating were unmelted Mo and W fragments (point 1 and Point 2). The crystal structure was mainly composed of Ni–Cu solid solution and Ni_x_Si_y_ phase (point 4). The enrichment of Si at grain boundaries of the Ni–Cu–10(Mo,W,Si) coatings (point 3) was similar to that of the Ni–Cu–10(W,Si) coatings. White (point 5) phases were inferred to be WSi_2_, (Mo,W)Si_2_. Infinite solid solutions of the (Mo,W)Si_2_ mixed phase were generated from WSi_2_ and MoSi_2_ due to their similar crystalline structure. Grey phases (point 6) were inferred to be MoSi_2_ and (Mo,W)Si_2_. Mo_5_Si_3_ may also exist in grey phases because of insufficient Si contents.

Elemental distribution within the Ni–Cu–15(Mo,W,Si) coating was further analyzed with an EPMA. Elemental face scanning images are depicted in Figure 6. Bright white grains are unmelted W grains. Unmelted Mo is not clear in Figure 6a. There are clear mutations in Ni, Cu, and Si contents between crystalline grains and grain boundaries, with higher Ni and Si contents at the grain boundaries. In addition, the positions of Mo and W at the grain boundaries are essentially the same but the content of Mo is higher than that of W. There are small apparent enrichments of Mo, Ni, and Si around the W particles, as shown in zone I of Figure 6. It follows that the MoNiSi phase formed around W. It was also observed that W particles were surrounded by Mo. The coexistence of elemental Mo, W, and Si was apparent in zone II, with the grey phases being dominated by Mo and the light grey phases dominated by W. The dispersion of elemental Mo is higher than that of elemental W as compared with Figure 6d,e. According to the results obtained using the EPMA, there are interdiffusion regions around the MoW particles, forming the M(Mo/W)NiSi phase. The coexistence of elemental Mo, W, and Si proves that (Mo,W)Si_2_ exists at grain boundaries. The grain size around the Mo–W–Si compound is smaller.

The time for atoms to diffuse decreased sharply and the rate of grain growth was restricted when subjected to cooling [35]. The temperature gradient at the bottom of the coating was smaller than at the top because of residual heat in the substrate. The rate of cooling was proportional to the temperature gradient, meaning that the size of the grain decreased from the bottom to the top of the coating. Temperature gradient (G) and growth rate (R) jointly dominate the solidification of coating microstructures, where G/R determines the morphology of solidification grains [36,37]. The trend in grain morphology is large grain–cellular crystal–dendrite crystal (from the bottom to the top of the coating).

On the one hand, silicides, which form because NiMoW elements easily react with Si, first appear as nuclei in nucleation. Silicides, represented by Ni_x_Si_y_ and MSi_2_ (where M is Mo,W), provide nucleation sites and promote nonhomogeneous nucleation. On the other hand, grain growth can be restricted by the addition of elemental Mo, W, and Si. The grain restriction factor (GRF) is defined as $Q=m_{L}C_{O}\left(K-1\right)$, where $m_{L}$ is the slope of the liquidus line, $C_{O}$ is the concentration of alloying elements, and K is the partition coefficient [38]. Components with low melting points are enriched at the front of the liquid–solid interface because components with a high melting point initially solidify. Initial solute concentrations at the front of the liquid–solid interface increase, contributing to the increase in the GRF. The grain size is inversely proportional to the GRF; thus, the growth in grain restriction is obvious after the addition of elemental Mo, W, and Si [39].

Therefore, the crystal structure of coatings is regulated by the precipitation phase, resulting in varying degrees of grain refinement, as shown in Figure 7. The Ni–Cu coating is relatively dense, and crystals are relatively homogeneous, with no significant elemental segregation. The Ni–Cu–10(W,Si) coating is characterized by the presence of Ni_x_Si_y_ and WSi_2_ silicides at grain boundaries, which effectively inhibit grain growth, resulting in a slight refinement. The presence of elemental Mo can refine the grain of the coating, which is consistent with the results obtained by Shaikh and Zhang [40,41]. The Gibbs free energy of Mo_5_Si_3_ is lower than that of Ni_2_Si and W_5_Si_3_, on the basis of Figure 3. Mo captures some Si atoms and first appears as small nuclei in the process of nucleation. These nuclei provide locations for nucleation ahead of the W_5_Si_3_, accelerating the nucleation process. Thus, significant grain refinement occurs in Ni–Cu–10(Mo,W,Si) with (Mo,W)Si_2_ phases compared with the Ni–Cu–10(W,Si) coating and the WSi_2_ phases. Higher silicide contents exist in the Ni–Cu–15(Mo,W,Si) coating, which forms a homogeneous internal organization and phase dispersion, resulting in the greatest degree of grain refinement.

### 3.3. Micro-Hardness

The cross-sectional micro-hardness distribution of coatings is shown in Figure 8. The average micro-hardness values of all four coatings were 137 HV, 391 HV, 402 HV, and 428 HV, respectively, whereas the micro-hardness value of the Cu–Cr–Zr substrate was only 71 HV. The Ni–Cu–15(Mo,W,Si) coating displayed the best performance, with a micro-hardness of 428 HV, 590% greater than the Cu–Cr–Zr substrate. There was a certain transition zone area due to the dilution effect of the laser cladding process.

Increases in micro-hardness were found in the Ni–Cu coating due to the internal formation of the Ni–Cu infinite solid solution. Ni_x_Si_y_, WSi_2_ (Mo,W)Si_2_, and MoNiSi were observed after the addition of Mo, W, and Si. The existence of silicides plays an important role in improving strength due to strengthening the grain boundary and precipitation. According to the Hall–Petch relationship, more effective obstruction of dislocation movement is found when the number of grain boundaries increases [42]. Diffuse precipitated phases act as pins, impeding dislocation movement and further enhancing hardness [43]. Slight improvements in the Ni–Cu–10 (Mo,W,Si) coating occurred compared with the Ni–Cu–10(W,Si) coating. This is because the grains were finer in the Ni–Cu–10 (Mo,W,Si) coating. In addition, MoSi_2_ and WSi_2_ with the same Cb11-type crystal formed an infinite solid solution, strengthening the toughness of the MoSi_2_ coating [44]. The micro-hardness of the Ni–Cu–15(Mo,W,Si) coating was slightly higher than that of Ni–Cu–10(Mo,W,Si), due to the difference in silicide content. Increased numbers of diffuse-reinforced particles can further refine grain.

### 3.4. Wear Properties

The friction coefficients of the substrate and coatings are shown in Figure 9. The average friction coefficients for the Cu–Cr–Zr substrate, Ni–Cu, Ni–Cu–10(W,Si), Ni–Cu–10(Mo,W,Si), and Ni–Cu–15(Mo,W,Si) coatings were 0.358, 0.409, 0.440, 0.443, and 0.473, respectively (Figure 9a). The substrate and Ni–Cu coating fluctuated during the stable wear stage after the running-in stage. However, fluctuations in the friction coefficient curves for Ni–Cu–10(W,Si), Ni–Cu–10(Mo,W,Si), and Ni–Cu–15(Mo,W,Si) coatings were smaller than in the stable stage. Comparing the friction coefficient curves, the coefficient of friction of the Ni–Cu–15(Mo,W,Si) coating was the largest, despite its most stable fluctuation.

The friction coefficient increased with a decrease in hardness. Shear behavior occurred in the attached soft metal under normal and parallel forces, and the soft phase was consistently coated on the friction pair; therefore, the coefficient of friction was low. The Ni–Cu coating with a Ni–Cu solid solution demonstrated a high micro-hardness, which slightly increased the friction resistance. In coatings with silicide, friction coefficients increase. The reason is that a large number of in situ hard silicide particles disperse in the coatings and play a role in the dispersion and fine grain strengthening effects, which improve friction resistance during the grinding process of friction pairs [45,46].

The wear rates of the substrate and coatings are depicted in Figure 9b, where the wear rate of the substrate is 128.1 mg/km. Wear rates of the Ni–Cu, Ni–Cu–10(W,Si), Ni–Cu–10(Mo,W,Si), and Ni–Cu–15(Mo,W,Si) coatings decreased sequentially. In addition, the trend in wear rate was consistent with the micro-hardness results. The wear rate of the Ni–Cu–15(Mo,W,Si) coating was 34.7 mg/km, 27% greater than that of the Cu–Cr–Zr substrate.

To further investigate the wear properties, the wear track morphology was measured using an ultra-deep field view device. Figure 10 shows the wear track morphology and wear scar cross-sectional profile of the specimen. The substrate wear depth color variations are evident in Figure 10a. The Ni–Cu, Ni–Cu–10(W,Si), Ni–Cu–10(Mo,W,Si), and Ni–Cu–15(Mo,W,Si) coatings exhibited improvements in wear depth compared with the substrate, as shown in Figure 10b–e. Cross-sectional profiles of the abrasion scars on the substrate and coatings are shown in Figure 10f. The wear scars on the substrate exhibited certain warpage and were large in area and depth, with a depth of 225 μm. The most significant improvement in wear scars was found in the Ni–Cu–15(Mo,W,Si) coating, with a wear scar depth of 50 μm, 0.2 times the size of that of the substrate. Furthermore, the wear scars on the substrate were not only warped but also exhibited a large area and depth. The Ni–Cu coatings demonstrated a small improvement in wear resistance after solid solution strengthening. The Ni–Cu–10(Mo,W,Si) and Ni–Cu–15(Mo,W,Si) coatings depicted good wear resistance after dispersion strengthening and fine grain strengthening, followed by the Ni–Cu–10(W,Si) coating.

To investigate the wear mechanisms of the substrate and coating, the micromorphology of the wear scars and debris was analyzed; the results of the mill mark enlargement SEM and EDS are depicted in Figure 11 and Figure 12, respectively. The substrate is warped at the edges of the wear scars in Figure 11a, with large flakes of abrasive chips. The magnification of abrasion scars is shown in Figure 12a, where there is significant detachment and flaking within the substrate, which is attributed to the deformation. The substrate was severely abraded by adhesive wear. Elemental analysis showed there was no elemental Fe at wear scars, suggesting that the actual wear process was between the substrates. Figure 11b shows that slight warping occurred at the edges of abrasion scars of Ni–Cu coating. The abrasive chips were in the form of flakes and chunks. An enlargement of wear scars of the Ni–Cu coating is shown in Figure 12b, with a small amount of detachment and localized fine-scale abrasion scars. A small amount of elemental Fe is present in wear scars of the Ni–Cu coating, whereas high contents of elemental O are observed on the surface. The wear of the Ni–Cu coating is mainly adhesive wear and oxidative wear. SEM images of wear scars and debris of the Ni–Cu–10(W,Si) coating are presented in Figure 11c, with localized areas of W-phase-rich smearing. The abrasive chips were mainly finely broken chunks. Deformation of the hardness phases is observed in the ovals in Figure 11c. An enlarged view of the abrasion scars of the Ni–Cu–10(W,Si) coating is shown in Figure 12c. The furrows are more pronounced with a small amount of detachment from adhesive wear. Elemental Fe and O in the wear scars of the Ni–Cu–10(W,Si) coating indicate that some oxidative wear is present. In short, the wear mechanism of the Ni–Cu–10(W,Si) coating is dominated by adhesive wear and abrasive wear, in addition to some oxidative wear. The shape of the abrasion scars of the Ni–Cu–10(Mo,W,Si) and Ni–Cu–15(Mo,W,Si) coatings are shown in Figure 11d,e, respectively. Some plowings and detachment are present in the Ni–Cu–10(Mo,W,Si) coating, while smear is also observed in the Ni–Cu–15(Mo,W,Si) coating, except for detachment. Abrasive chips of the Ni–Cu–15(Mo,W,Si) coating are fine, compared with other coatings (Figure 11a–e). The results of the EDS analysis show that there are elemental O and Fe on the abrasive surfaces, with higher contents of elemental O on the surface of the Ni–Cu–15(Mo,W,Si) coating (Figure 12e). Slight adhesive and oxidative wear were observed in both coatings; however, the abrasive wear of plastic deformation plays a decisive role.

When the substrate was ground against GCr15 grinding material, flakes were eventually formed and the edges of the grinding tracks warped significantly. The substrate was softer than the grinding material, adhering the copper alloy to the grinding material and fracturing continuously under higher normal loading forces. The Ni–Cu coating displayed better wear resistance than the Cu-Zr-Cr substrate, which was attributed to the high content of the solid solution. There was higher shear strength of the internally formed Ni–Cu solid solution compared with Cu–Cr–Zr substrate. The Ni–Cu–10(W,Si) coating exhibited increased wear resistance due to the effect of the precipitated phase WSi_2_ and fine grain reinforcement. The Ni–Cu–Mo–W–Si coatings included Mo_5_Si_3_, MoSi_2_, WSi_2_, and MoNiSi silicides. MoSi_2_ and WSi_2_ exhibited high hardness and elastic modulus values, which increases the energy required for crack expansion during chip formation [47,48]. The Ni–Cu–15(Mo,W,Si) coating showed a relatively high content, thus demonstrating excellent wear resistance. Abrasive particles were pressed into the surface by a normal force and pushed forward by the tangential force of abrasive particles. The silicides were finely crystalline, toughened, and diffusely strengthened, which improved the micro-hardness. The harder the phase, the more resistant it is to plastic deformation caused by the ground material. The areas with fewer reinforced phases eventually formed abrasive chips due to repeated plastic deformation under abrasive particles.

## 4. Conclusions

Four types of coatings were prepared on copper alloys surface by coaxial powder-feeding laser cladding technology. The microstructure, micro-hardness, and frictional wear properties of the coatings were investigated. The main conclusions are presented subsequently.

The existence of silicides, such as Ni_x_Si_y_, WSi_2_ (Mo,W)Si_2_, and MoNiSi, plays an important role in grain refinement due to the restriction of grain growth and the promotion of nucleation. Significant refinement of grain occurs with a Ni–Cu–10(Mo,W,Si) coating compared with a Ni–Cu–10(W,Si) coating because Mo_5_Si_3_ preferentially precipitates and accelerates the nucleation process. The most significant grain refinement in the Ni–Cu–15(Mo,W,Si) coating was attributed to the increase in precipitated phases. The crystalline size of the coatings changed from fine to large crystalline structure (from the top to the bottom of the coatings).

Fine grain strengthening, solid solution strengthening, and dispersion enhancement perform important roles in improving the micro-hardness of coatings. Meanwhile, the micro-hardness of silicide-containing coatings is much greater than that of a Ni–Cu coating and Cu–Cr–Zr substrate. This is attributed to assorted and varied silicides which pin the grain boundaries and hinder the dislocation movement. The Ni–Cu–15(Mo,W,Si) coating displayed the best performance, with a micro-hardness of 428 HV, 590% greater than that of the Cu–Cr–Zr substrate.

The wear resistance of coatings improved, compared with the Cu–Cr–Zr substrate. As a consequence of the diffuse distribution of the silicide and the fine grain strengthening effect, the Ni–Cu–15(Mo,W,Si) coating showed excellent wear resistance, with a wear rate of 36.49 mg/km, approximately 27.1% greater than the substrate. The wear mechanism transformed from adhesive wear in the Ni–Cu coating with no silicides to abrasive wear in the Ni–Cu–15(Mo,W,Si) coating with a high level of silicides.

## Figures and Tables

**Figure 1 materials-16-00284-f001:**
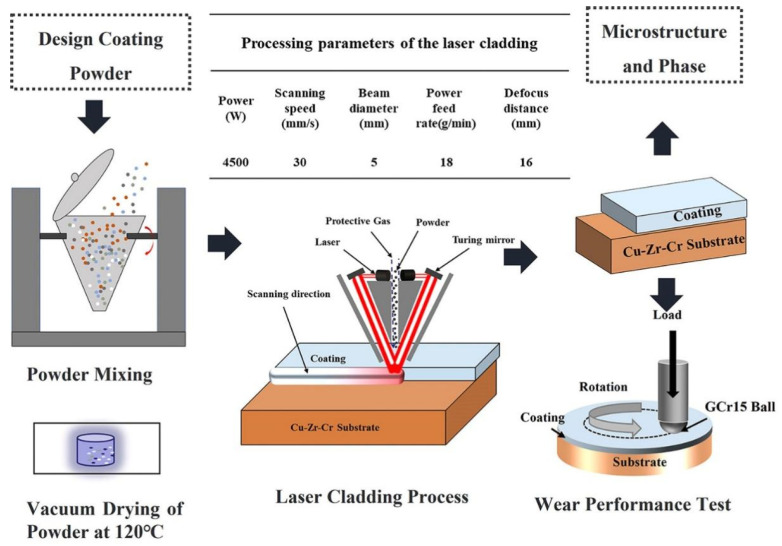
Preparation process, cladding process parameters, and performance test of coatings.

**Figure 2 materials-16-00284-f002:**
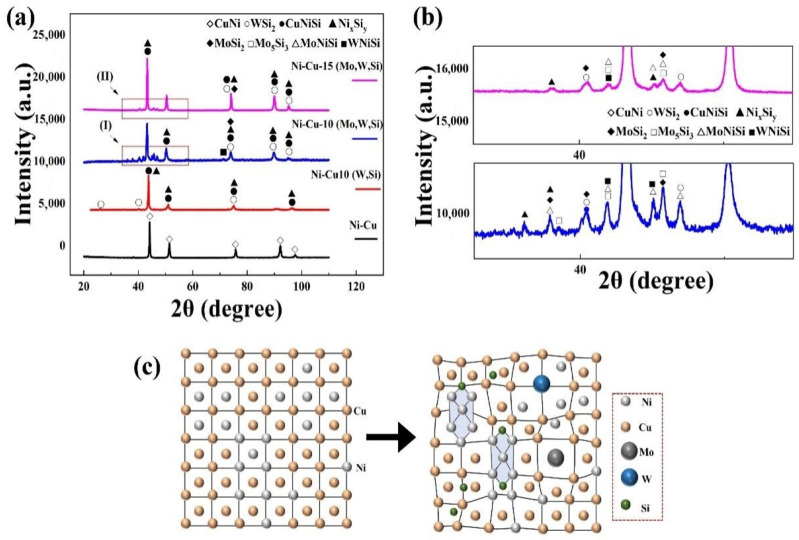
(**a**) XRD patterns of different coatings; (**b**) Enlarged XRD patterns of (I) and (II); and (**c**) schematic of lattice distortion.

**Figure 3 materials-16-00284-f003:**
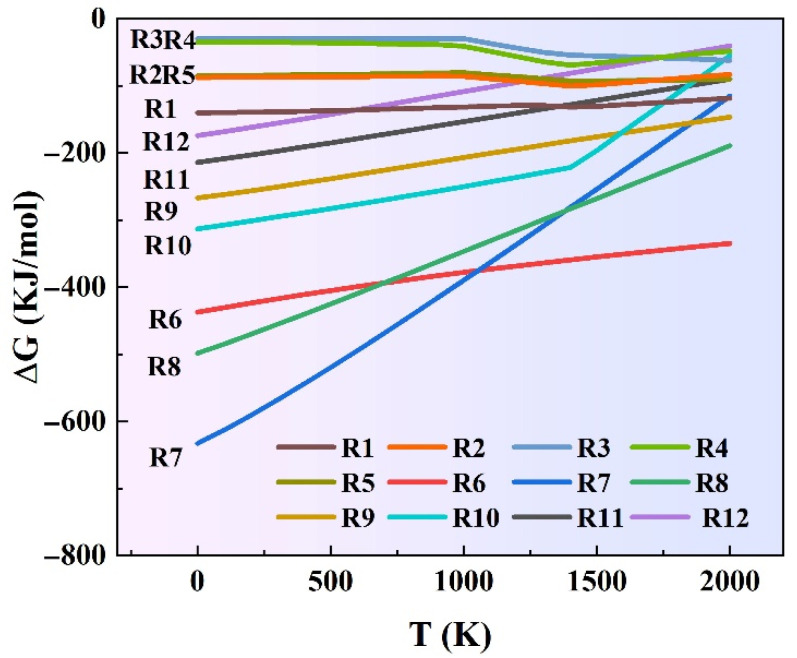
Gibbs free energy (ΔG) of Reactions (R1)–(R12).

**Figure 4 materials-16-00284-f004:**
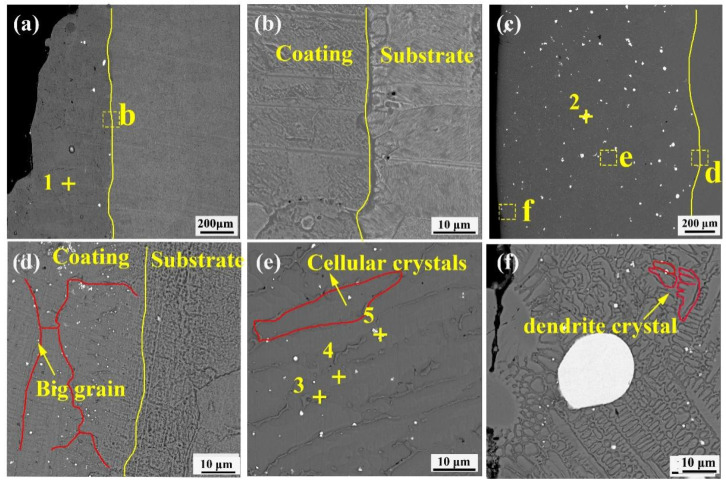
Cross-sectional morphology of the Ni–Cu coating and Ni–Cu–10(W,Si): (**a**) cross-sectional morphology of the Ni–Cu coating; (**b**) bonding interface between the Ni–Cu coating and the substrate; (**c**) cross-sectional morphology of the Ni–Cu–10(W,Si) coating; (**d**) bonding interface between the Ni–Cu–10(W,Si) coating and the substrate; (**e**) structure of the middle area of the Ni–Cu–10(W,Si) coating; (**f**) morphology of the upper part of the Ni–Cu–10(W,Si) coating.

**Figure 5 materials-16-00284-f005:**
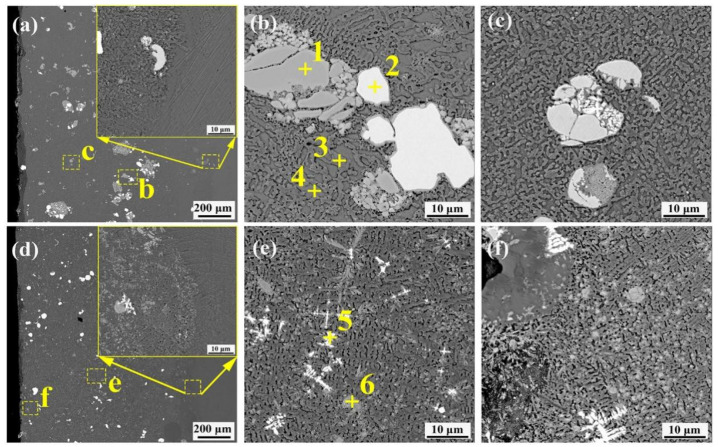
Cross-sectional morphology of the Ni–Cu–10(Mo,W,Si) and Ni–Cu–15(Mo,W,Si) coatings: (**a**) cross-sectional morphology of the Ni–Cu–10(Mo,W,Si) coating; (**b**) structure of the upper area of the Ni–Cu–10(Mo,W,Si) coating; (**c**) structure of the middle area of the Ni–Cu–10(Mo,W,Si) coating; (**d**) cross-sectional morphology of the Ni–Cu–15(Mo,W,Si) coating; (**e**) structure of the middle area of the Ni–Cu–15(Mo,W,Si) coating; (**f**) structure of the upper area of the Ni–Cu–15(Mo,W,Si) coating.

**Figure 6 materials-16-00284-f006:**
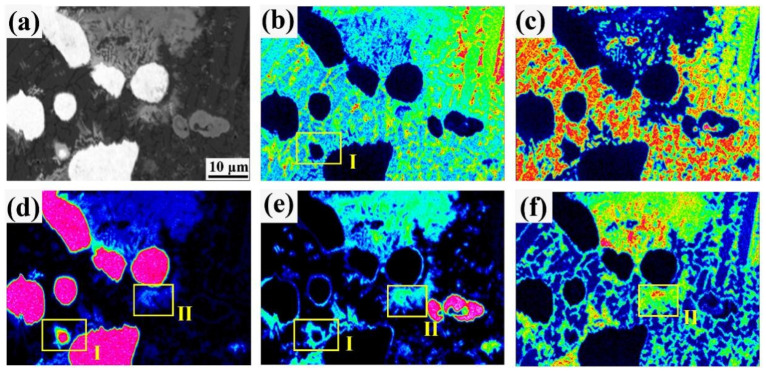
Elemental distribution surface scans of the Ni–Cu–15(Mo,W,Si) coating: (**a**) BSE graph of the Ni–Cu–15(Mo,W,Si) coating; (**b**) Ni; (**c**) Cu; (**d**) W; (**e**) Mo; (**f**) Si.

**Figure 7 materials-16-00284-f007:**
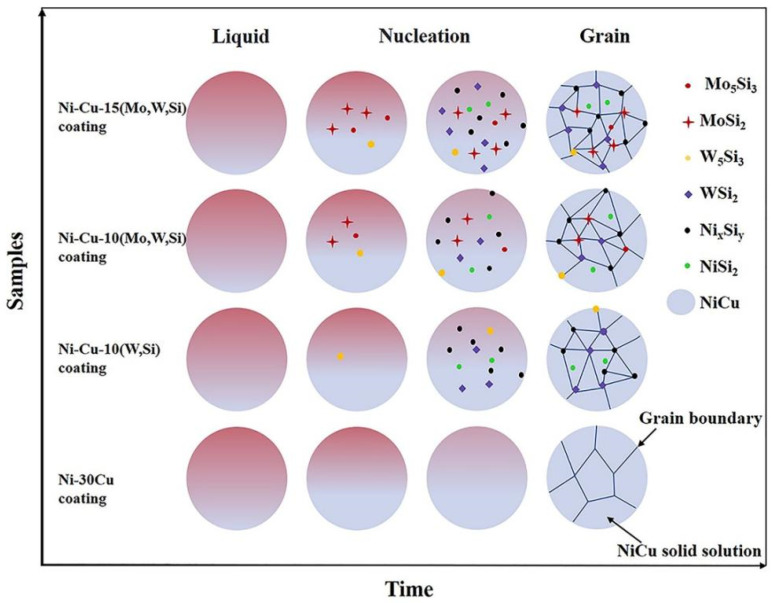
Simplified diagram of the organization of four coatings.

**Figure 8 materials-16-00284-f008:**
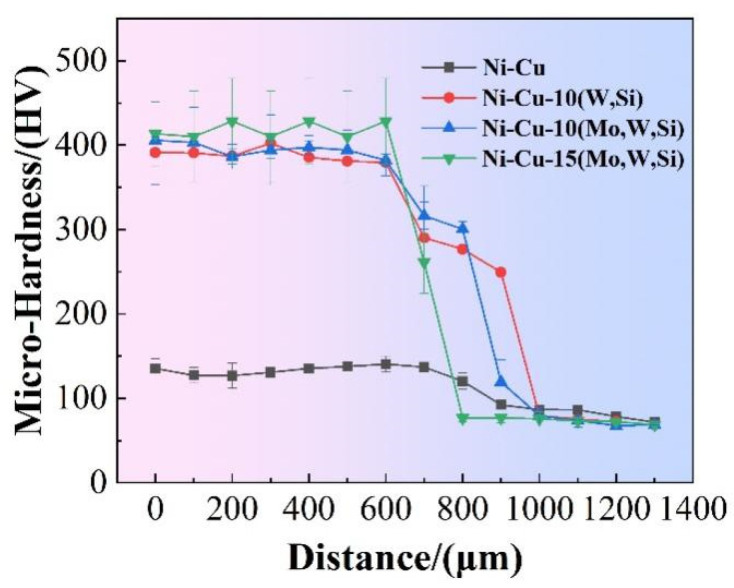
Micro-hardness graphs of four coatings.

**Figure 9 materials-16-00284-f009:**
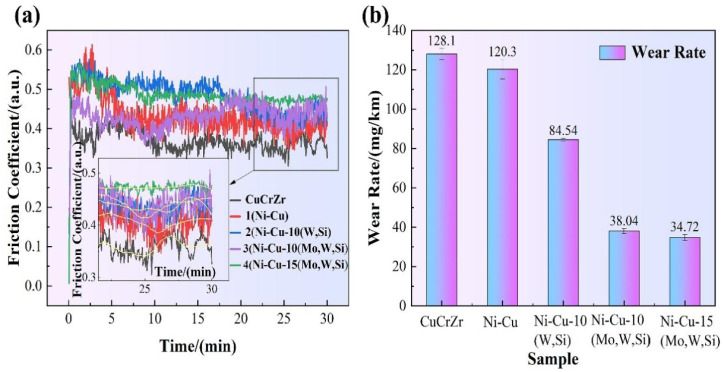
(**a**) Friction coefficients of the substrate and coatings, and (**b**) wear rates of the substrates and coatings.

**Figure 10 materials-16-00284-f010:**
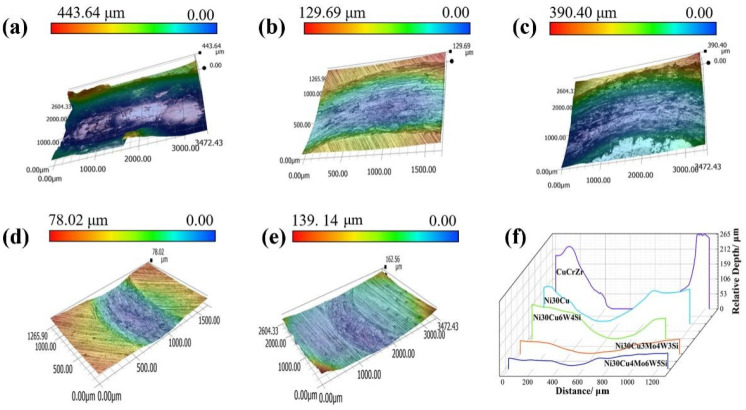
Wear surface morphology: (**a**) Cu–Cr–Zr substrate; (**b**) Ni–Cu coating; (**c**) Ni–Cu–10(W,Si) coating; (**d**) Ni–Cu–10(Mo,W,Si) coating; (**e**) Ni–Cu–15(Mo,W,Si) coating; (**f**) profile of the wear scar section.

**Figure 11 materials-16-00284-f011:**
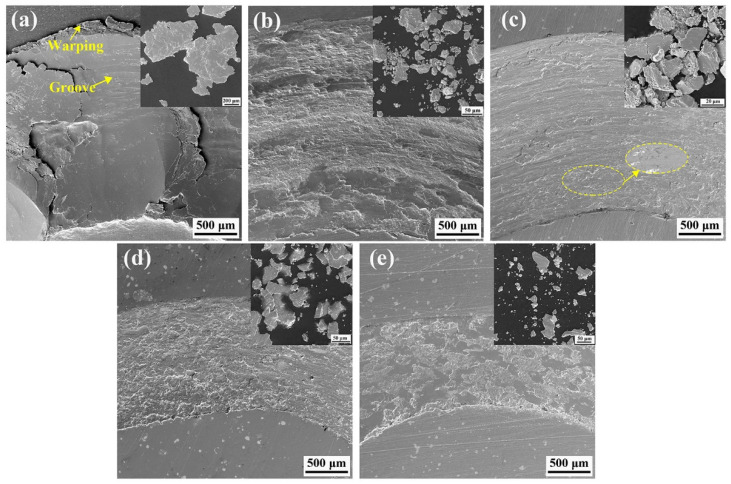
Wear scars and debris: (**a**) Cu–Cr–Zr substrate; (**b**) Ni–Cu coating; (**c**) Ni–Cu–10(W,Si) coating; (**d**) Ni–Cu–10(Mo,W,Si) coating; (**e**) Ni–Cu–15(Mo,W,Si) coating.

**Figure 12 materials-16-00284-f012:**
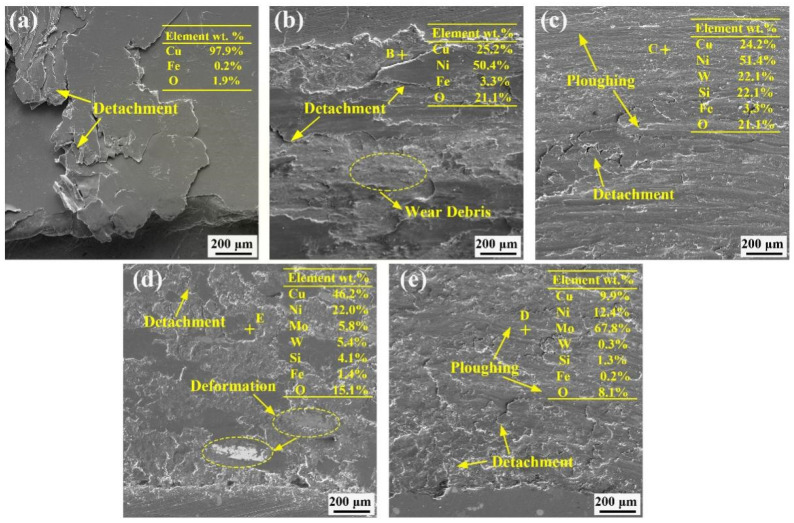
The micromorphology wear scars: (**a**) Cu–Cr–Zr substrate; (**b**) Ni–Cu coating; (**c**) Ni–Cu–10(W,Si) coating; (**d**) Ni–Cu–10(Mo,W,Si) coating; (**e**) Ni–Cu–15(Mo,W,Si) coating.

**Table 1 materials-16-00284-t001:** Coating powder mass components (wt.%).

Powder	Ni	Cu	Mo	W	Si
Ni–Cu	70	30	-	-	-
Ni–Cu–10(W,Si)	65	30	-	6	4
Ni–Cu–10(Mo,W,Si)	60	30	2.5	4.6	2.9
Ni–Cu–15(Mo,W,Si)	55	30	3.7	7.0	4.3

**Table 2 materials-16-00284-t002:** EDS elemental (at.%) composition of the coatings shown in Figure 4.

	Ni	Cu	W	Si
1	70.4	29.6	-	-
2	2.6	-	97.3	-
3	35.3	59.4	0.2	5.1
4	35.7	61.2	0.3	2.8
5	30.7	23.7	12.4	33.2

**Table 3 materials-16-00284-t003:** EDS elemental composition (at.%) of the coatings shown in Figure 5.

	Ni	Cu	Mo	W	Si
1	2.8	6.5	87.4	0.1	3.2
2	1.0	-	1.1	97.8	-
3	55.9	27.3	2.3	0.7	13.8
4	45.0	49.0	-	0.4	5.6
5	13.3	2.8	4.9	23.3	49.7
6	14.2	2.4	27.9	2.6	52.9

## Data Availability

The data presented in this study are available on reasonable request from the corresponding author.

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
