# Peer review of "Microstructure and Wear Resistance of Laser-Clad Ni–Cu–Mo–W–Si Coatings on a Cu–Cr–Zr Alloy"

_materials, 2022, doi:10.3390/ma16010284_

Round 1

Reviewer 1 Report

The manuscript reports the results of an interesting investigation on the microstructure and wear resistance of four laser clad materials on a Cu-Cr-Zr alloy substrate. I believe that manuscript can be accepted for publication after the following corrections and modifications.

1- There are some typo and syntax errors in the manuscript.

2- Some small changes in the title is suggested: “Microstructure and wear resistance of laser clad Ni-Cu-Mo-W-Si coatings on a Cu-Cr-Zr alloy”

3- Line 73: Please explain how the mixing parameters may affect or ensure the sphericity of the used powders?!

4- Line 120: Please elaborate on what is meant by “sub-stable phase”.

5- Stoichiometry of reaction 4 is not correct.

6- Fig. 3 is too crowded. Can the lines be labeled/numbered?

7- Fig. 4-f (and lines 159 and 232): I believe that the marked grains are not “equiaxed crystals”. In fact, they are dendrite arms observed as isolated crystals in 2D metallographic cross sections!

8- I could not understand the following sentences. Please rephrase:

Lines 171-173: “The reason why … polarized”.  

 Lines 234-236: “When the nucleation … accelerate the nucleation process”

9- Lines 199-209: Please discuss and explain the reasons for the results presented.

10- Lines 17-218: It is claimed that decrease in the grain size is due to reduced G/R ratio. However, it is a common knowledge now that grain size is controlled by G.R. In fact, G/R controls the morphology of the grains not their size!

11- Fig. 7 needs to be modified to represent the microstructure and grain structures more precisely. For example, isn’t there any grains in the CuNi coating? Grains in the other 3 coatings look mostly like a mess rather than an assortment of different grain morphologies! ‘Organization” does not fit the title of the figure.

12- In discussing the microstructural effects of different elements, the authors are suggested to make use of the ‘grain restriction factor (GRF)” concept for each element.

13- Line 244: please report the hardness values without decimal point.

14- Line 287: Please correct the “friction sub”.

15- Can the authors suggest any change in the processing parameters to avoid the unmelted W and Mo powders!

16- Please explain on how these unmelted particles may affect the wear resistance and wear mechanisms of their corresponding samples.

Author Response

We would like to express our thanks for the reviewer’s constructive and very helpful comments, which are attached below. Corrections and modifications are highlighted in yellow in the manuscript. 

Reviewer 2 Report

Dear Authors!

English style and grammar should undergo professional major revision 

The choice of coaxial powder feeding laser cladding technology should be justified.

The advantages of using the silicides should be explained in thr text.

Line 64-65. Add the table with the substrate properties, or list them in the textt

Line 66. After wire cutting, about 1 mm of the material should be removed to avoid the effects on its composition and properties. Did You do this?

Line 82. Give the reference if the results are published.

Line 84. What was the thickness of the coating(s)?

Line 90-91. Specify the name and issue of the XRD analyzing software, and the database used for it.

Line 101-102. Please, explain why did you selected 100 and 400 micrometers? It is unclear, as far ase You did not disclosed the thickness of the coating

Line 112, formula 1 misses the time of the experiment. If to multiply 2*?*?*?, You will not get the total friction path, You should also multiply this by the test duration in minutes. Please, modify the formula.

Line 113-114. Friction test parameters should be described one-by-one, but not as a crowd.

Figure 2, ab. Remove pink background

Figure 4f. Line 166. It contains significant porocity (black areas). Please, describe it and conclude about its effect on the coating performance.

Line 181. ''Light-white'' - this has no sense. Mayby, just light?

Line  183-187. Whan do You mean: upper structure, middle structure? What is that? On Fig. 5a there is a clearly different surface layer (left side of the picture). What is that? Please, describe and explain it.

Line 236-238. Based on which data You made a conclusion about the strengthening of grain boudaties? For this, I think, the nanoindentation test is mandatory.

Figure 8. Why the Ni-Cu-10 (Mo,W,Si) coating is twice thinner than the others?

Line 269. The CuCrZr alloy composition should be added to table 1, and its microhardness should be added to Fig. 8. Also, somwhere, describe its microstructure.

Lines 276-286. In Your case, I think, the increase of coefficient of friction is caused by the increase in hard silicide particles (which also increase the hardness). These particles act as (both fixed and free) abbrasive between clamped surfaces during the friction. Actually, not the copper decreases friction coefficient, but hard abbrasive silicide particles increease it. Grain refinement also imparts this. This is a reason. This part should be reexplained correctly

Figure 11. The magnification on the right-up corner inserts can not be read. Please, enlarge the font.

Figure 12. Please, explain the difference between delamination and tetachement.

Figure 12c. Please, explain what we should see inside the ovals?

Line 351-352. Cr and Zr also produce solid solution strengthening of Cu. I think, the explanation shold be changed.

The style of references should be changed according to Materials requirements

Author Response

(The authors gave the same response as above.)

Round 2

Reviewer 2 Report

Dear Authors.

Thank You for the work done, especially for the additional explanations and tests. Now, the manuscript may be published